# Characterization, Classification, and Authentication of *Polygonatum sibiricum* Samples by Volatile Profiles and Flavor Properties

**DOI:** 10.3390/molecules27010025

**Published:** 2021-12-21

**Authors:** Xile Cheng, Hongyuan Ji, Xiang Cheng, Dongmei Wang, Tianshi Li, Kun Ren, Shouhe Qu, Yingni Pan, Xiaoqiu Liu

**Affiliations:** 1School of Chinese Materia Medica, Shenyang Pharmaceutical University, Shenyang 110016, China; believele@126.com (X.C.); jhy129413@163.com (H.J.); renkun_spu@163.com (K.R.); shouhequ@126.com (S.Q.); panyingni@163.com (Y.P.); 2School of Pharmacy, Bozhou Vocational and Technical College, Bozhou 236000, China; xiangcheng_bzzy@126.com; 3School of Pharmacy, Shenyang Pharmaceutical University, Shenyang 110016, China; 4Bozhou Yonggang Pieces Factory Co., Ltd., Bozhou 236000, China; lixlixiang@163.com

**Keywords:** *Polygonatum sibiricum*, electronic nose, electronic tongue, gas chromatography-mass spectroscopy, chemometric analysis

## Abstract

The importance of monitoring key aroma compounds as food characteristics to solve sample classification and authentication is increasing. The rhizome of *Polygonatum sibiricum* (PR, Huangjing in Chinese) has great potential to serve as an ingredient of functional foods owing to its tonic effect and flavor properties. In this study, we aimed to characterize and classify PR samples obtained from different processing levels through their volatile profiles and flavor properties by using electronic nose, electronic tongue, and headspace gas chromatography-mass spectrometry. Nine flavor indicators (four odor indicators and five taste indicators) had a strong influence on the classification ability, and a total of 54 volatile compounds were identified in all samples. The traditional Chinese processing method significantly decreased the contents of aldehydes and alkanes, while more ketones, nitrogen heterocycles, alcohols, terpenoids, sulfides, and furans/pyrans were generated in the processing cycle. The results confirmed the potential applicability of volatile profiles and flavor properties for classification of PR samples, and this study provided new insights for determining the processing level in food and pharmaceutical industries based on samples with specific flavor characteristics.

## 1. Introduction

In recent years, there has been a growing interest in the medicinal and edible plants as functional foods and health-promoting dietary supplements with remarkable market value. *Polygonatum sibiricum* Red., a type of traditional medicinal herb and edible plant, is mainly distributed in the northern parts of China (such as Liaoning, Hebei, and Inner Mongolia Provinces), North Korea, and Mongolia [1]. Historically, the rhizome of *Polygonatum sibiricum* (PR, huangjing in Chinese) was documented as “got the essence of soil, be the first in tonics” in the Compendium of Materia Medica (Ming Dynasty, 1590 A.D.). Huangjing is one of the most popular traditional medicinal herbs with a wide range of beneficial effects such as replenishing Qi, nourishing Yin, fortifying spleens, moistening lungs, and tonifying kidneys [2]. Thus, it has been considered to have various pharmacological effects, including promoting physical energy, improving gastrointestinal function, protecting respiratory system, improving sexual performance, and strengthening the immune system. Furthermore, PR can also be used to treat some symptoms such as fatigue, weakness, indigestion, inappetence, diabetes, cough, sexual dysfunction, backache, knee pain, and premature hair greying [3].Therefore, huangjing is frequently used in Chinese people’s daily diets [4], and a variety of functional foods with huangjing as main raw material are supplied on the market [5].

However, huangjing must be processed through traditional Chinese processing technology to be used in traditional medicine and the food industry in China. The steaming technology, an effective method to change food from raw to mature, plays an important role in the fields of food manufacturing and food processing worldwide. It is well known that steaming is a traditional processing method for huangjing, especially the strategy of nine cycles of steaming and drying, which accounts for 56.6% of all ancient literatures on the processing methods of huangjing [6]. The steaming process can affect major quality parameters associated with food products, such as color, odor, taste, polysaccharides, volatile compounds, and secondary metabolites [7,8,9,10].

As a sort of popular tonic food in China, PR is often processed by repeated steaming and drying procedures until the rhizome turns black, soft, and sweet, which can significantly enhance its nourishing function and avoid throat irritation. Over the years, methods for determining the effective processing level of PR have been explored, while the steaming time, processing degree, and objective judgment indexes (e.g., color, texture, taste, and odor) of PR are still unclear. In addition, there are few reports on evaluations of the sensory quality of PR processed under different numbers of processing cycles. 

Over the past few years, gas chromatography-mass spectroscopy (GC-MS), electronic nose (E–nose), and electronic tongue (E–tongue) have been increasingly applied in the food industry for quality control [10,11,12,13]. E–nose and E–tongue have provided support for the objective expression of the odor and taste of medicinal and edible homologous medicines. Many volatile compounds have been found to have several therapeutic properties, including antioxidation, antitumor, anti-inflammation, and antimicrobial activities [14]. Three terpenoids, i.e., α-pinene, camphene, and β-caryophyllene, have been found to have the potential to treat a variety of inflammatory diseases such as respiratory inflammation, atopic dermatitis, arthritis, and neuroinflammation [15]. Geraniol has been found to possess various pharmacological properties, including antitumor, antioxidant, anti-inflammatory, antimicrobial, antidiabetic, cardioprotective, and neuroprotective activities [16,17]. Beta-caryophyllene has been identified a CB_2_ receptor agonist with pharmacological activities such as antidiabetic, anti-inflammatory, and anticancer [18]. Various techniques have been used to extract volatile compounds in the field of herbal medicine, such as steam distillation, simultaneous distillation–extraction, static headspace, and headspace solid-phase microextraction (HS-SPME) [19,20,21]. In contrast to a liquid–liquid extraction method, static headspace is a non-destructive and non-invasive method that avoids the contamination of solvent impurities. As compared with HS-SPME, the static headspace is simple and low cost [22]. Therefore, static headspace is suitable for routine analysis and meets the criteria of green analytical chemistry principles. GC-MS can be used to reveal changes in volatile components during processing. Chemometric analysis is a simple strategy to identify traditional Chinese medicines, which has been widely used in quality control and evaluation of Chinese materia medica [23,24]. The chemometric methods of principal component analysis (PCA) and hierarchical cluster analysis (HCA) are commonly used in the field of herbal medicines for identification investigations, including species, geographical location, processing production, and harvesting time, as well as other factors that influence the quality of herbal materials [25]. GC-MS combined with chemometric analysis was recently used to analyze ginseng [26], goji berry [27], rhizomes of Curcuma [28], and Ephedrae herba [29]. Thus, modern analytical techniques integrated with chemometrics are becoming useful analytical tools for quality control of herbal medicines.

The objective of this study was to understand the influence of traditional Chinese processing on volatile profiles and flavor properties of PR samples. E–nose, E–tongue, and headspace GC-MS were applied to characterize volatile constituents and flavor properties in the nine processing cycles of PR. A chemometric analysis was conducted with the aim to accurately distinguish PR with different processing levels, and to screen out potential markers for the identification of processing production. The results are useful for monitoring the changes in sensory quality and identifying the processing level of PR samples.

## 2. Results and Discussion

### 2.1. pH Measurement

The color of the samples gradually became darker after each processing cycle (Appendix A). Figure 1 shows that the pH value for PR samples decreased progressively as the processing level increased, undergoing dynamic changes from 5.50 (initial value) to 3.91. The decrease in pH could be due to the progression of the Maillard reaction and the generation of organic acids accompanied by a reduction in amino groups [30]. 

### 2.2. Electronic Nose

#### 2.2.1. Electronic Nose Response to Different Processing Levels of PR

The typical E–nose responses to different processing levels is shown in the Appendix A (Appendix A). A total of 10 PR samples with different processing levels were detected by E–nose, and the response curve is shown in Appendix A. The horizontal axis is the sampling time (s), and the vertical axis is the response signal value of the E–nose sensor (G/G0 or G0/G). Each curve represents the change in response intensity of a sensor during the sampling time. The response values conductivity ratio of seven sensors were G/G0, and the response values of the other three sensors, i.e., W1C, W3C, and W5C were G0/G, in which G is the conductivity of the sensor after contacting the sample gas, and G0 is the conductivity of sensor cleaned by standard activated carbon filtered gas. The curve represents the response signal value of 10 metal oxide sensors to the odor of the sample (different colors represent different metal oxide sensors).

The data show that the response values of the 10 sensors of the E–nose gradually increased with an increase in acquisition time, and the curve gradually stabilized at 30 s. When the change range of the curve was small, the sensitivity of the electronic nose to the odor had reached a stable state. As shown in Appendix A, stabilization was reached within 50–60 s. The W1W sensor was the most sensitive variable, and the W1W response value of the RPR sample was relatively low. With increased processing, the W1W response value gradually increased, and the fifth steamed sample (PPR5) stabilized.

#### 2.2.2. Data Analysis on Odor Characteristics of PR with Different Processing Levels

According to the odor response value of different samples, an odor characteristic map of PR samples was established, which is shown in Figure 2. The circumference represents the name of the sensor, and the response value of different samples is displayed in polar coordinates by color patches. By using a one-way ANOVA test and Kruskal–Wallis test (Appendix A), there were differences in the 10 aroma indicators of PR with different processing levels (*p* < 0.05). According to Figure 2 and sensor response values (Appendix A), sensors W1W, W5S, W2W, and W1S were more sensitive to the odor of samples, and their response values varied significantly from the RPR sample to the PPR9 sample. The odor indicators of the 10 samples were different, which was further combined with the PCA unsupervised analysis method to analyze the odor characteristics of samples collected by E–nose to obtain the significant odor characteristics of PR samples with different processing levels.

#### 2.2.3. Differentiation of Processed PR Samples by PCA

To evaluate the influence of the steaming process on the grouping of PR, a PCA was performed on the dataset of the response values of E–nose samples. The biplots of score and loading of the processed samples are presented in Figure 3. The data showed that the grouping of the samples was from left to right along the t1 axis with increased processing levels. The RPR sample, first PR sample (PPR1), second PR sample (PPR2), and third PR sample (PPR3) were located on the left side of the score plot, while the fourth PR sample (PPR4), fifth PR sample (PPR5), sixth PR sample (PPR6), seventh PR sample (PPR7), eighth PR sample (PPR8), and ninth PR sample (PPR9) were located on the right side of the score plot, among which there was an overlap in the PPR5–PPR9 samples. It was preliminarily considered that the smell of the fifth PR sample was similar to that of the ninth PR sample.

The cumulative variance contribution R2X of the first three principal components reached 0.997, and the predictive ability parameter Q2 was 0.984, which indicated that the PCA model was a well-fitted and predictive model. Principal component factor loading analysis showed the contribution rate of each factor variable to the principal component. The greater the absolute value of the load eigenvector, the greater the contribution to the principal component. The principal component factor loading matrix is presented in Table 1. The first principal component (PC1) had the largest amount of information, and the independent variance explanation contribution reached 0.984. The absolute values of the eigenvectors of sensors W1W (0.596), W1S (0.379), and W5S (0.375) were larger than the remaining sensors. In the second principal component (PC2), the absolute values of the eigenvectors of sensors W2W (0.610) and W1C (−0.307) were relatively large. In addition, in the third principal component (PC3), the absolute values of the eigenvectors of sensors W3S (0.882) and W6S (0.304) were relatively large. Therefore, the PC1 mainly depended on sensors W1W, W1S, and W5S; the PC2 depended on sensors W2W and W1C; and the PC3 depended on sensors W3S and W6S.

Figure 3B shows that the factor loading corresponding to the characteristic values of different sensors was mainly distributed along the p1 axis on the right side of the loading plot, and some overlap was observed. Sensors W1W, W2W, W1S, and W5S significantly contributed to the differentiation of PR samples. The characteristic values of sensors with high overlap were highly similar. In combination with Table 1, the absolute values of the eigenvectors of the three partially overlapped sensors W3C, W5C, and W2S were small, causing little contribution to the grouping of the samples, and therefore redundant information could be eliminated.

The results of the PCA analysis showed that the number of steaming and drying cycles had a significant effect on the odor characteristics of PR samples. The significant odor differences between the RPR sample and the PR samples were W1W, W2W, W1S, and W5S. After the fifth processing cycle, the odor index basically did not change, and the samples from PPR5 to PPR9 were pooled.

### 2.3. Electronic Tongue

#### 2.3.1. Data Analysis on Taste Characteristics of Different Processed PR Samples

According to the taste response value of different samples, the taste characteristic map of PR samples is presented in Figure 4. The characteristic map of E–tongue was similar to that of E–nose, which presented the taste difference of the samples. Using the one-way ANOVA test and Kruskal–Wallis test, it is shown in Appendix A that there were differences in the nine taste indicators of PR with different processing levels (*p* < 0.05). According to the data presented in Figure 4 and taste sensor response values (Appendix A), the taste differences of 10 different samples were mainly reflected in the indexes of sourness, sweetness, astringency, aftertaste-astringency, umami, and richness. With an increased degree of processing, the sourness index increased significantly. However, the sweetness decreased, the astringency and aftertaste-astringency increased, the umami taste decreased, and the richness increased.

#### 2.3.2. Differentiation in Processed PR Samples by PCA

To distinguish PR samples from different processing levels, a PCA was also performed on the E–tongue dataset, taking the response values of taste sensors as input variables. Figure 5A shows that the RPR, PPR1, PPR2, PPR3, and PPR4 samples were located on the left side of the score plot, whereas the PPR5, PPR6, PPR7, PPR8, and PPR9 samples were located on the right side of the score plot, in which there was an overlap in the PPR5–PPR8 samples. It was preliminarily considered that the taste from the fifth PR sample to the eighth PR sample was similar.

The cumulative variance contribution R2X of the first three principal components reached 0.956, and the predictive ability parameter Q2 was 0.847, which indicated that the PCA model had good fitting and prediction properties. The taste index corresponding to each sensor of the E–tongue was regarded to be a factor variable, and the loading analysis of principal component factor was performed. The principal component factor loading matrix is shown in Table 2. The PC1 gave the most information; the independent variance explanation contribution reached 0.682, and the absolute values of the eigenvectors of the taste indexes for sourness (0.391), astringency (0.390), sweetness (−0.390), richness (0.382), umami (−0.376) were large. The taste indexes for saltiness (−0.579) and aftertaste-A (−0.438) in the second principal component had large eigenvector absolute values, and the taste indexes for bitterness (0.772) and aftertaste-B (0.443) in the third principal component had high absolute values. The data showed that the PC1 mainly depended on the taste indexes for sourness, astringency, sweetness, richness, and umami, the second principal component depended on the taste indexes for saltiness and aftertaste-A, and the third principal component depended on the taste indexes for bitterness and aftertaste-B.

Figure 5B shows that the taste indexes for sourness, richness, astringency, sweetness, and umami had a greater contribution to the classification of PR samples with different processing levels. Moreover, Appendix A shows that the taste characteristics of RPR were mainly sweet and umami, and the taste characteristics of the samples from PPR5 to PPR9 were sourness, richness, and astringency.

The results of the PCA analysis showed that the number of steaming and drying cycles had a significant effect on the taste characteristics of PR samples. Furthermore, the taste index did not significantly change after the fifth processing cycle.

The minimum detectable value of the taste sensor of the SA402B electronic tongue was a 20% change in concentration, and the response output value could be converted into “taste information”, thereby, indicating that one unit of the response value of the taste sensor represented the taste difference caused by a 20% concentration difference in the sample, which was also the smallest unit of taste difference that people began to perceive. The full range of taste perception is 25 units, and the range of taste response that was experienced is acid (−13–12), bitter (0–25), astringency (0–25), salty (−6–19), and sweet (0–25) [31]. Table 3 shows that among the nine taste indicators, the sweetness values of the 10 samples of PR with different processing levels ranged from 8.62 to 15.15, which accounted for a large portion of the sweetness range perceived by humans; only the sour taste value of the ninth PR sample (PPR9) was in the range of human perception, which was experienced. In addition, other taste indexes accounted for a small proportion of human perception. Therefore, what people really experienced before and after the steaming process was that there was always sweetness, and there was sourness only in the ninth PR sample.

Through literature studies, we found that the sample solution of traditional Chinese medicine measured by electronic tongue included water-extracted solutions [32,33]. In Chinese clinical application, PR is mostly taken after water decoction. For the PR samples measured by the E–tongue, to achieve taste that was closer to clinical application, the experimental samples were prepared using a water extraction method.

### 2.4. Headspace GC-MS

#### 2.4.1. Volatile Compounds in the Different Samples by Headspace GC-MS

To evaluate the effect of the traditional steaming process on the flavor of PR samples, HS-GC-MS was used to analyze the volatile compounds of each sample afforded by different processing levels. The total ion chromatograms of all samples with different processing levels are shown in Figure 6A. The main peak appeared at 5.3 min, and its ionic strength decreased significantly with an increase in the number of processing cycles. Two compounds, hexanal and octane, were identified from the main peak. Hexanal had a lower odor threshold as compared with octane. It showed that n-hexanal could be used as a component marker to distinguish samples with different processing levels. The individual chromatograms of 10 samples are shown in Appendix A. The number and relative content of volatile compounds from each sample are shown in Figure 6, and additional details are presented in Appendix A.

A total number of 54 volatile compounds were detected in all PR samples, which were classified as follows: 4 alcohols, 4 phenols, 7 aldehydes, 9 terpenoids, 4 ketones, 1 ester, 12 furans/pyrans, 1 carboxylic acid, 5 alkanes/alkenes, 4 nitrogen heterocycles, 1 amine, and 2 sulfides. The most abundantly found flavor families were furans/pyrans, terpenoids, and aldehydes (Figure 6B).

The relative contents of the main volatile compounds in RPR in decreasing order were as follows: aldehydes (35.56%) > furans/pyrans (22.60%) > alkanes/alkenes (22.16%) > ketones (6.45%). After the steaming and drying cycle, the relative contents of volatile compounds in the PPR9 sample were furans/pyrans (78.76%) > ketones (7.81%) > nitrogen heterocycles (5.90%) > aldehydes (3.30%) > alkanes/alkenes (1.73%). These findings showed that as compared with the RPR sample, the total relative content of furans/pyrans was significantly increased, while aldehydes and alkanes/alkenes were decreased. The relative content of furans/pyrans in the PPR9 sample reached 78.76%, which increased by 56.16% as compared with that of the RPR sample (22.60%). The total relative contents of aldehydes and alkanes/alkenes in the PPR9 sample were 3.30% and 1.73%, respectively, which decreased by 32.26% and 20.43% as compared with that in the RPR sample (35.56% and 22.16%). With an increase in the number of cycles, alcohols, terpenoids, ketones, sulfides, and nitrogen heterocycles first increased, then decreased, and then stabilized, but no significant changes were observed for esters, carboxylic acids, and phenols (Figure 6C).

As shown in Appendix A, there were 26 common compounds in all samples. With an increase in processing cycles, common volatile compounds were significantly decreased. As compared with RPR, 15 compounds were newly produced in processed samples (shown in Appendix A), including 6 furans (2,5-dimethylfuran, furfural, 2,5-furandione, 2-furanmethanol, 2-ethyl-5-methylfuran, and 5-methylfurfural), 3 terpenoids (geraniol, (+)-alpha-muurolene, and beta-caryophyllene), 2 phenols (m-cresol and butylated hydroxytoluene), 1 ketone (cyclohexanone), 1 aldehyde (2-ethyl-2-hexenal), 1 nitrogen heterocycles (1,4-dimethylpyrazole), and 1 sulfide (dimethyl trisulfide). These data were in accordance with data presented in the literature [34,35,36]. As shown in Appendix A, 13 volatile components could not be gradually detected, of which 3 chemical components were unique to RPR, namely 3-methyl-1-pentanol, undecane, and gamma-undecalactone. The difference between the contents and types of volatile compounds present determine the unique aroma profile of each sample. 

In previous studies, it has been shown that aldehydes were relatively abundant headspace volatiles and had a very low odor threshold, therefore, they played a significant role in the characteristic flavor of food [37]. Aldehydes were found to be the major class of volatile compounds in all samples. Most aldehydes have been reported to be highly associated with almond, green, and citrus aromas [8,38]. Hexanal made a significant contribution to the aldehydes in RPR (29.88%). As a flavor active compound, hexanal conferred a rancid smell at high amounts, while in low content it produced a pleasant green aroma [39]. Hexanal decreased as the processing cycle increased, whereas nonanal (citrus, green), benzaldehyde (almond aroma), and beta-cyclocitral (minty) increased.

The content of alkanes/alkenes significantly decreased with increased processing cycles. The RPR sample had the highest ratio of alkanes/alkenes, with the most abundant octane, which accounted for 21.09% of the alkanes/alkenes class. Generally, alkanes/alkenes did not significantly contribute to the aroma due to their high odor threshold and they had an odor similar to gasoline [40].

The content trend of ketone showed that it increased greatly and rapidly in the early stage, and then decreased and stabilized in the later stage. The occurrence of ketone compounds usually endowed food with a cheese, fruit, and caramel flavor [41].

The alcohols detected in all samples were mainly 1-octen-3-ol, 2-ethylhexanol, and 2-methylcyclohexanol, which commonly produced mushroom, earthy, green, rose, and citrus aromas. The PPR1 sample reached the highest alcohol content of 11.90%, with the most abundant 1-octen-3-ol, which was responsible for the mushroom-like odor.

Furan/pyran is a class of oxygenated heterocyclic compounds for food flavors. Because the furan ring is more stable than the pyran ring, more furan is produced during the pyrolysis process. Furans were detected in all samples, which were mainly produced during Maillard reactions and thermal degradation of sugars [42]. The processed samples contained abundant furans, which increased rapidly with increased processing cycles. Among furans, furfural was the dominant compound in the post-processing stage, which contributed to the flavors of almond, bread, burnt, and spicy in food. 

RPR had the characteristics of a “long-time smell of raw flavor, and a dazzling feeling” [34]. However, hexanal has been shown to be the main component of indoor environmental irritants, and was irritating to eyes, the respiratory system, and the skin [43,44]. The presence of undecane can also cause damage to the central nervous system, respiratory irritation, and even chemical pneumonitis [45]. The content of hexanal was the highest in RPR, and undecane was a unique component in RPR. Hexanal and undecane might be volatile components that can cause eye irritation, but further experiments are needed to prove this. 

According to the literature [46,47], key-aroma compounds have been identified through calculation of their relative odor activity values (ROAV). Compounds with ROAV ≥ 1 were considered to be key flavor components that significantly contributed to the final aroma profile, and compounds with 0.1≤ ROAV < 1 had an important modifying effect on the overall flavor of the sample. All the compounds with ROAV ≥ 0.1 are listed in Table 3. Thus, a total of 16 compounds were found to effectively contribute to the final flavor profile of the PR samples. Among them, four volatiles were observed to be key volatiles (ROAV ≥ 1) in the RPR and PPR1 samples, six volatiles were key volatiles in the PPR2 sample, and nine volatiles were key volatiles in the PPR3 and PPR4 samples. In the PPR5–PPR9 samples, 10 volatiles were key volatiles. The data showed that repeated steaming and drying was conducive to the diversity of the flavor of PR samples. The types of key flavor compounds did not significantly change after the fifth processing cycle. 

The heatmap of key flavor components of PR samples with different processing levels is shown in Figure 7. Analysis of the relative amounts of 16 key volatiles in each processing level showed that the concentration of hexanal, 1-octen-3-ol, and 2-pentylfuran were significantly high in all the PR samples. The PPR1 and PPR2 samples showed a similar or improved composition of volatile compounds as compared with the RPR sample. Furthermore, the PPR3 and PPR4 samples exhibited a similar flavor chemical composition. The samples from PPR5 to PPR9 also exhibited similar compositions, indicating the flavor reached a steady state after four cycles.

In general, volatile markers associated with grassy, green, gasoline-like, and irritation, (e.g., hexanal and undecane) decreased during the processing cycle, whereas thermal-load, cabbage (e.g., dimethyl disulfide and dimethyl trisulfide), browning, almond, and burnt (e.g., furfural) indicator compounds increased during the processing cycle. Due to the abovementioned findings, PPR was more suitable for food applications with desirable aromas than RPR. It was useful for the determination of processing level in food and pharmaceutical industries based on samples with specific flavor characteristics.

#### 2.4.2. Differentiation of Ten Different Processed PR Samples by PCA

A PCA was performed on the data matrix (30 samples × 12 volatile classes) to observe a possible sample distribution according to different processing levels. The cumulative variance contribution R2X of the first three principal components reached 0.971, and the predictive ability parameter Q2 was 0.836. As shown in the score plot of PCA (Figure 8A), the samples exhibited a tendency to form three major groups. The RPR and PPR1 samples were all located in the third quadrant of the graph, while the PPR2, PPR3, and PPR4 samples were located in the second quadrant. The PPR5 and PPR6 samples were distributed close to the fourth quadrant, whereas the PPR7, PPR8 and PPR9 samples were located in the fourth quadrant of the graph. According to the PCA loading plot (Figure 8B), aldehydes, alkanes/alkenes, ketones, nitrogen heterocycles, and furans/pyrans had a greater influence on the differentiation ability as compared with other components. The aldehydes and alkanes/alkenes mainly distinguished RPR and PPR1 samples from other processed samples. In addition, ketones and nitrogen heterocycles contributed more to the PPR3 and PPR4 samples. Finally, the contents of furans/pyrans correlated more with the PPR5, PPR6, PPR7, PPR8, and PPR9 samples. 

### 2.5. Correlation Analysis of pH and E–Tongue

A Pearson correlation analysis was conducted to determine the relationship between pH and E–tongue sensors (Figure 9A). The pH value showed strong positive correlations with the sweetness and umami of the RPR sample, while pH value showed strong negative correlations with the sourness and richness of the PPR9 sample. The result indicated that pH value could discriminate PR samples by responding to human-perceived taste indicators such as sweetness and sourness.

### 2.6. Correlation Analysis of E–Nose and E–Tongue

The flavor of PR samples was developed through a series of complex processes. Taste was affected by the perception of odor, and its changes could be partially explained by changes in smell. Figure 9B shows that E–nose sensors were associated with E–tongue sensors, and changes in the taste would be further reflected in the changes of odor. Moreover, it was revealed that E–nose had the feasibility of evaluating taste-presenting substances and achieving rapid and non-destructive prediction of sample taste. 

### 2.7. Correlation Analysis of E–Nose and GC-MS

Sixteen volatile compounds with high relative odor activity values (ROAV ≥ 0.1) based on GC-MS analysis were selected to correlate with E–nose signals (Figure 9C). The results showed that the signal intensities of E–nose sensors had significant and positive correlations with the abundances of m-cresol, furfural, 2-furanmethanol, and 5-methylfurfural, which indicated that E–nose sensors were sensitive to furan derivatives and phenolic compounds. In comparison, the intensities of E–nose signals were negatively correlated with the abundances of hexanal, 2-heptanone, 1-octen-3-ol, and 2-pentylfuran. Therefore, these eight volatiles might be related significantly to the flavor of the PR samples. This showed that E–nose was capable of distinguishing PR samples by responding specifically to volatile compounds.

### 2.8. HCA of the Fusion Dataset of pH, E–Nose, E–Tongue, and GC-MS

A Cluster analysis was performed by using the between-groups linkage method and the squared Euclidean distance to the fused dataset comprised of pH, E–nose, E–tongue, and HS-GC-MS data. The dendrogram, shown in Figure 10, indicates that the relationships and distribution among the samples in different processing levels and four main clusters of the samples were as follows: RPR (Cluster 1); PPR1, PPR2, PPR3, and PPR4 (Cluster 2); PPR5, PPR6, PPR7, and PPR8 (Cluster 3); and PPR9 (Cluster 4). These classification groups were consistent with the PCA results in which all samples were fully distinguished according to their processing levels.

## 3. Materials and Methods

### 3.1. Samples and Sample Preparation

Fresh rhizomes of *Polygonatum sibiricum* (FPR) were harvested in October 2019 from Qingyuan County, Liaoning Province, one of the most famous PR growing regions in China; only fresh undamaged rhizomes with similar weight and size were selected for this study. There were 3 batches of samples, and the same batch of FPR was divided into 10 groups. According to the *Chinese Pharmacopoeia* (2020 edition), one group of FPR was processed into raw rhizomes of *Polygonatum sibiricum* (RPR), by boiling in water for 5 min, and then drying at 55 °C for 9 h. The remaining groups were subsequently processed into processed rhizomes of *Polygonatum sibiricum* samples (PPR1–PPR9), by 3 h of steaming (over boiling water), 1 h of simmering, followed by 9 h of oven drying (55 °C), repeated for 1–9 cycles.

### 3.2. pH Measurement

The pH value was determined using a Mettler Toledo FE 28 pH meter (Mettler Toledo Instruments Co. Ltd., Shanghai, China) at room temperature. Ten-gram sample thick slices (range 2–4 mm) were soaked in 10 times the volume of distilled water for 1 h at room temperature (25 °C), and then refluxed twice by boiling for 1.5 h each time. Each extract was filtered, and the combined extracts were used for pH measurement.

### 3.3. Electronic Nose Analysis

The E–nose analyses were performed using a commercial PEN3 electronic nose (Airsense Analytics GmbH., Schwerin, Germany). The instrument consists of a gas-rate control system, a sensor array, and a pattern analysis software (WinMuster, v.1.6., Airsense Analytics GmbH., Schwerin, Germany). The sensor array is composed of 10 metal-oxide semiconductor sensors, i.e., W1C, W5S, W3C, W6S, W5C, W1S, W1W, W2S, W2W, and W3S, which are sensitive to specific volatile compounds (Appendix A).

In brief, a sample of 4.0 g of powder (40 mesh) was put into a 250 mL beaker sealed with cling film and incubated for 30 min at 25 °C to reach the headspace equilibrium. The test parameters were set as follows: the flow rate was 200 mL/min, measurement time 60 s, and flush time 120 s. Responses of the sensors were expressed as the ratio of conductance G/G0 (G and G0, respectively, indicate the conductance of the sensors in contact with sample gas and clean gas), which varied with different substances. Each sample was measured in triplicate and the average of sensor responses after stabilization was taken for subsequent analysis. 

### 3.4. Electronic Tongue Analysis

The difference in taste of 10 samples was evaluated by using an electronic tongue (taste sensing system SA402B, Intelligent Sensor Technology Co., Kanagawa, Japan), which contained eight multichannel lipid membrane sensors: CA0, CT0, AAE, C00, AE1, GL1, AC0, and AE1(aftertaste), which were potentiometric sensors with specific sensitivity and selectivity to different taste substances. 

A total of 1 g of dried sample powder (40 mesh) was extracted with 100 mL distilled water under ultrasonication (40 kHz) for 30 min. Subsequently, the solution was centrifuged at 3000 rpm for 10 min at 25 °C and the supernatant was obtained. The supernatant of each sample measured 80 mL, was divided into two parts, and placed in two rows of circular sample cups in parallel. Acidity, bitterness, astringency, saltiness, sweetness, umami, and aftertaste were tested at room temperature (25 °C). Each sample was measured in quadruplicate. The data of the first cycle was removed, and the data of the following three cycles were taken as the measured values. After each measurement, the sensors were cleaned automatically.

### 3.5. Headspace GC-MS Analysis

The headspace autosampler combined with GC-MS was conducted to analyze the volatile compounds of PR samples. In brief, accurately weighed 3.0 g of different sample powder (40 mesh), was put into 20 mL headspace vials and capped with a PTFE/silicon septum. After incubation at 120 °C for 20 min, 10 µL of headspace gas was injected into to a gas chromatography inlet. The GC-MS analyses consisted of a TRACE 1300 Series GC and an ISQ Series MS (Thermo Fisher Scientific, Waltham, MA, USA). The volatiles were separated on a TG-5MS capillary column (30 m × 0.25 mm, 0.25 µm, Thermo Fisher Scientific, Waltham, MA, USA) based on the following analytical conditions: the injection port temperature was set to 250 °C, the flow rate of helium carrier gas was set to 1 mL/min, and the split ratio was 1:5. The condition of the oven temperature ramp was as follows: initial temperature of 35 °C for 2 min, 5 °C/min to 75 °C for 4 min, 15 °C/min to 95 °C for 1 min, and 20 °C/min to 150 °C, followed by 15 °C/min to 270 °C (maintained for 3 min). The programmed temperature mode was carried out in the range from 35 °C to 270 °C. Mass spectra were acquired in electron impact (EI) mode at 70 eV, with the ion source temperature of 280 °C, and a transfer line temperature of 280 °C. The detection was performed in full scan mode over a mass range of 40–250 *m*/*z*, with a scan time of 0.2 s per scan. 

First, the obtained GC/MS data were converted to an Analysis Base File (ABF) format using an ABF converter (https://www.reifycs.com/AbfConverter/index.html, accessed on 8 April 2021), and then imported into the MS-DIAL software for deconvolution, peak detection, alignment, and filtering. The identification of volatile compounds was performed using MoNA mass spectral libraries and Kovats retention index (RI), as shown in Appendix A. The results with dot and reverse dot product scores greater than 0.7 were selected. The retention indices of the identified compounds were calculated from the van den Dool and Kratz equation using the retention times of the alkane standards (C6–C20).The relative contents of components in each sample were determined by the normalization method [11,48]. 

### 3.6. Statistical Analysis

To differentiate and classify samples, in this study, unsupervised principal component analysis (PCA) was performed on the data, using SIMCA software (version 14.1, Umeå, Sweden). The statistical significance test was performed by one-way ANOVA and the Kruskal–Wallis test using SPSS 25.0 software (SPSS Inc., Chicago, IL, USA). For statistical testing, *p* < 0.05 was considered to be the critical level of significance. The hierarchical cluster analysis (HCA) was conducted using SPSS 25.0 software. The sensor response data were plotted to a polar coordinate heat map using the dycharts website (https://www.dycharts.com, accessed on 12 April 2021). The correlation analysis was performed using the OmicStudio tools, a free online platform for data analysis (https://www.omicstudio.cn/tool, accessed on 27 September 2021).

## 4. Conclusions

In this study, volatile profiles and flavor properties of PR samples prepared through traditional processing methods were characterized by E–nose, E–tongue, and HS-GC-MS. The total relative content of furans/pyrans increased significantly when the number of cycles increased, while the relative contents of aldehydes and alkanes/alkenes decreased. In addition, alcohols, terpenoids, ketones, sulfides, and nitrogen heterocycles exhibited a tendency of increasing first, and then decreasing and stabilizing. Esters, carboxylic acids, and phenols did not undergo significant changes. The Pearson correlation analysis indicated that the eight volatiles might be significantly correlated with the flavor of the PR samples. The PCA and HCA analysis showed that the number of cycles had a certain effect on the volatile compounds, odor, and taste of PR samples. The RPR sample was significantly different from other PR samples at different processing levels. It is worth noting that additional cycles had little effect on most features when the number of cycles exceeded four.

Therefore, processed PR after four cycles was considered to have similar volatile compounds and flavor characteristics based on the HCA results. The results confirmed the potential applicability of volatile profiles and flavor properties for classification of PR samples with different processing levels. The volatile fingerprint and classification of PR could allow for the selection of samples with specific flavor characteristics based on the food and pharmaceutical applications. Our results provide an effective way to determine the level of PR processing by monitoring the changes of volatile components and flavor characteristics. In conclusion, our study supported the application of this method for discrimination of processed PR samples, and also supported the establishment of similar methods for other traditional Chinese steaming products.

## Figures and Tables

**Figure 1 molecules-27-00025-f001:**
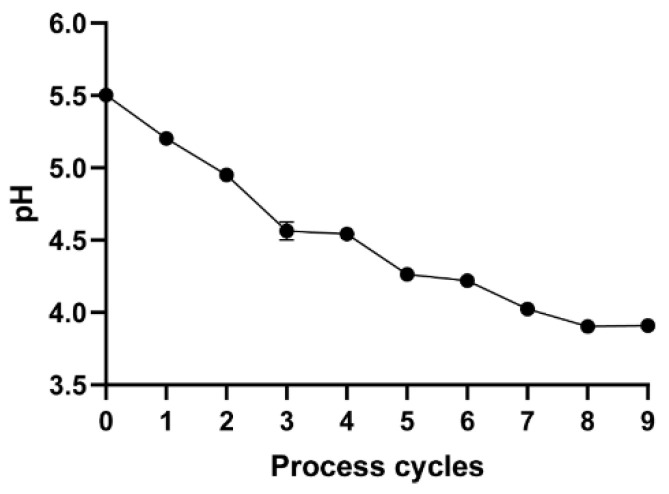
Changes in pH values for PR samples with different processing levels.

**Figure 2 molecules-27-00025-f002:**
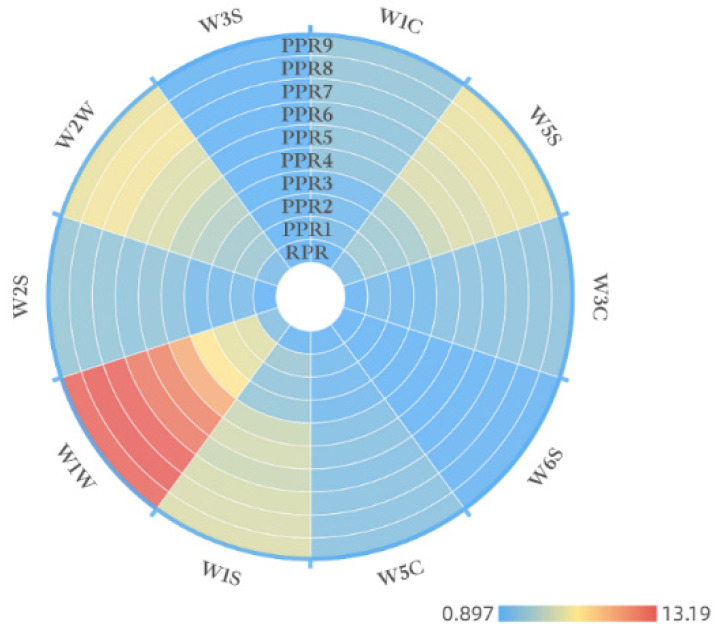
Odor characteristics of PR samples with different processing levels.

**Figure 3 molecules-27-00025-f003:**
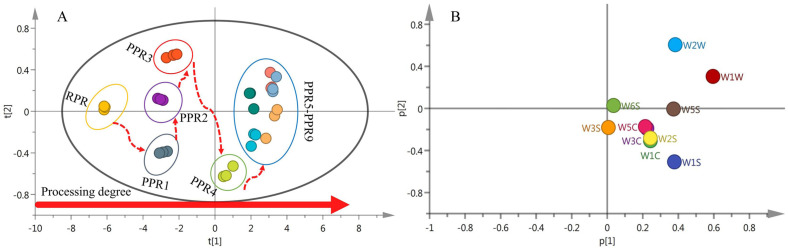
Principal component analysis PCA biplots for 10 different processed PR samples based on E–nose response data: (**A**) Score plot; (**B**) loading plot.

**Figure 4 molecules-27-00025-f004:**
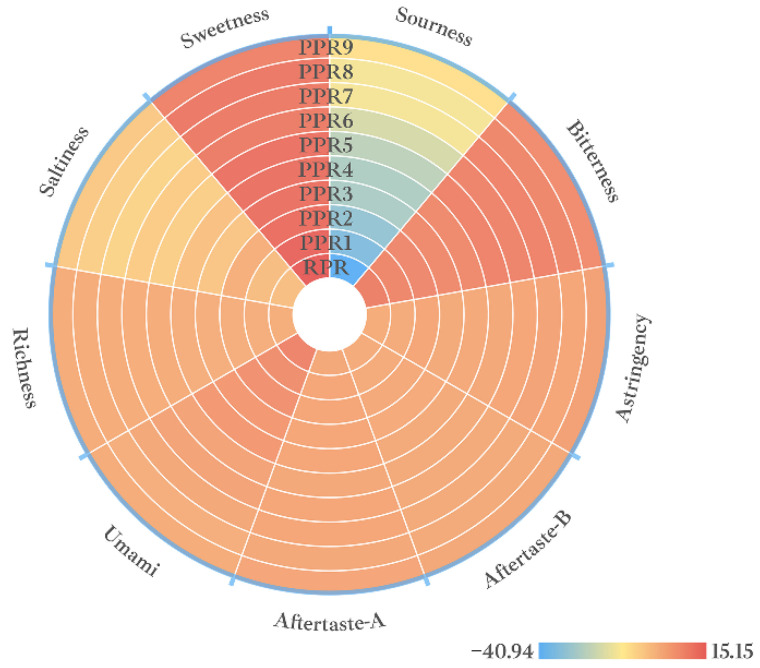
Taste characteristics of PR samples with different processing levels.

**Figure 5 molecules-27-00025-f005:**
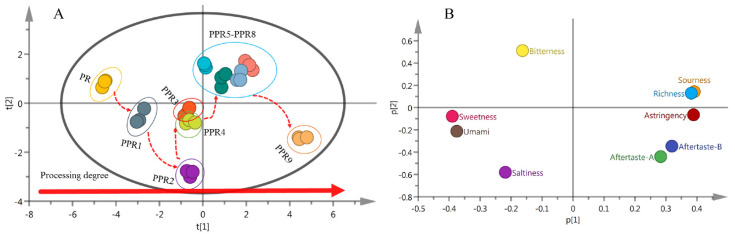
Principal component analysis PCA biplots for 10 different processed PR samples based on E–tongue response data: (**A**) Score plot; (**B**) loading plot.

**Figure 6 molecules-27-00025-f006:**
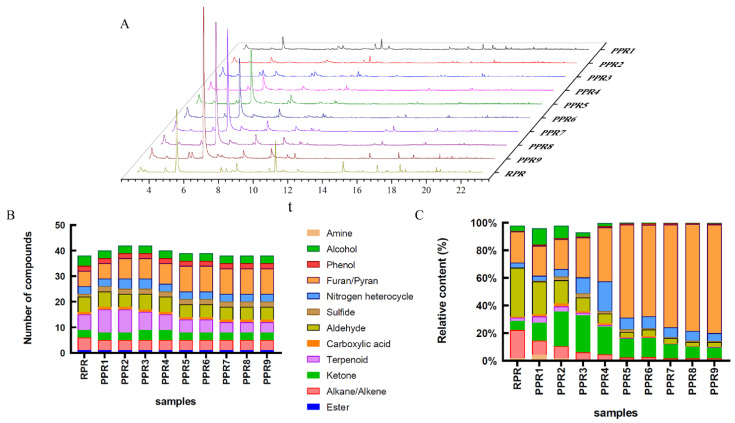
Volatile compounds of PR samples with different processing levels: (**A**) Total ion chromatograms; (**B**) number of volatile compounds; (**C**) relative content of volatile compounds.

**Figure 7 molecules-27-00025-f007:**
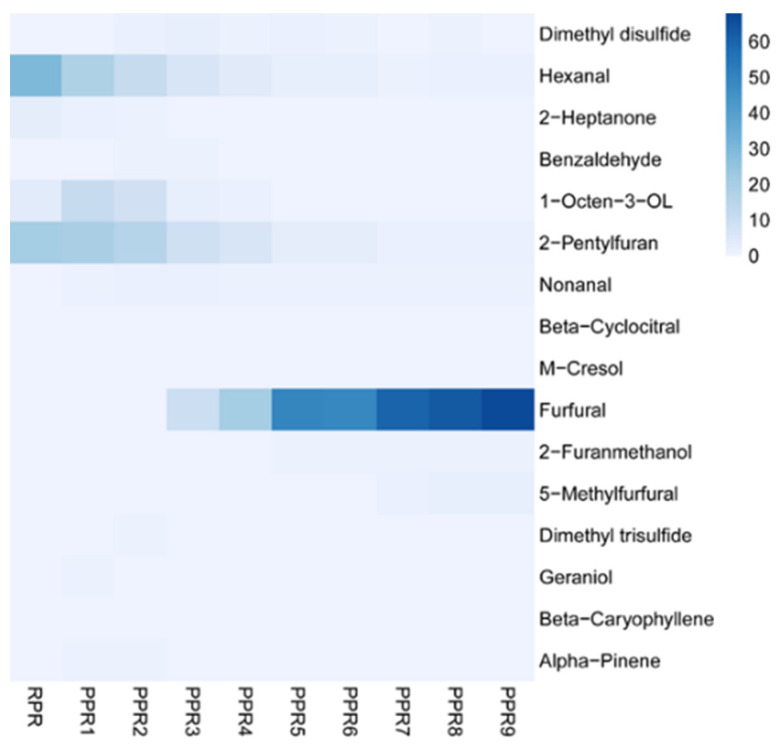
Heatmap of key flavor components of PR samples with different processing levels. The colors from white to blue represent the relative amounts of the flavor components from low to high.

**Figure 8 molecules-27-00025-f008:**
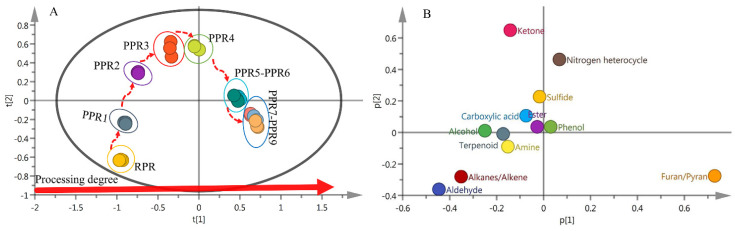
Principal component analysis PCA biplots for 10 different processed PR samples based on GC-MS data: (**A**) Score plot; (**B**) loading plot.

**Figure 9 molecules-27-00025-f009:**
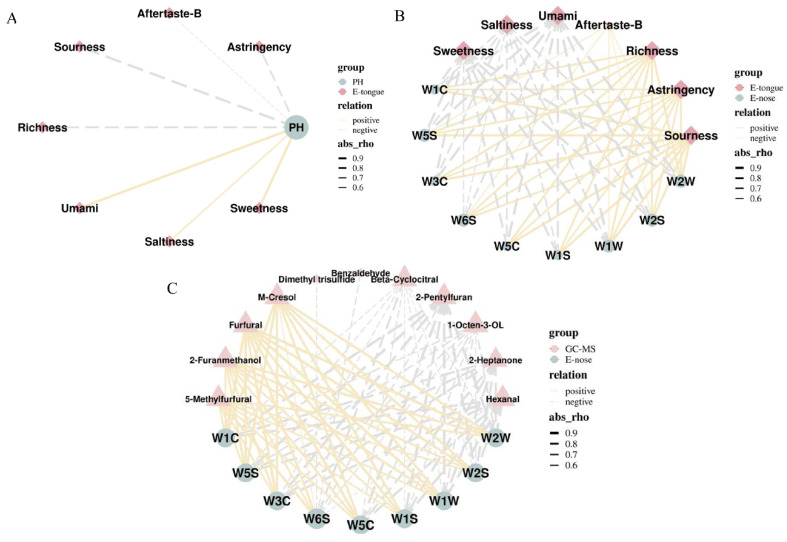
Correlations among pH, E–nose, E–tongue, GC-MS for 10 different processed PR samples: (**A**) Between pH value and E–tongue data; (**B**) between E–nose data and E–tongue data; (**C**) between E–nose data and GC-MS data.

**Figure 10 molecules-27-00025-f010:**
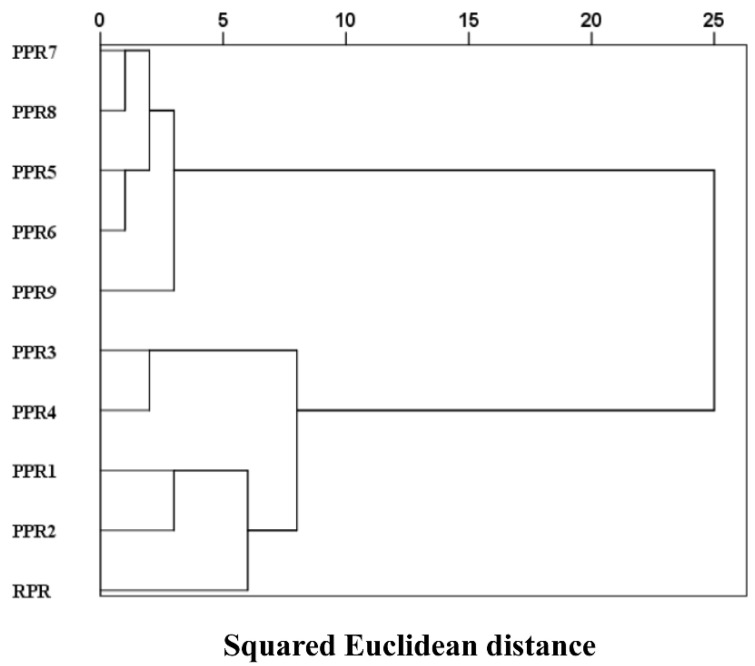
Dendrograms of HCA with a dataset including pH, E–nose, E–tongue, and GC-MS data.

**Table 1 molecules-27-00025-t001:** Principal component factor loading for E–nose data.

Sensor	PC1	PC2	PC3
W1C	0.242	−0.307	−0.059
W5S	0.375	−0.002	0.166
W3C	0.224	−0.192	−0.025
W6S	0.035	0.028	0.304
W5C	0.212	−0.172	0.004
W1S	0.379	−0.511	−0.290
W1W	0.596	0.308	0.054
W2S	0.245	−0.279	0.103
W2W	0.381	0.610	−0.014
W3S	0.005	−0.179	0.882
R2X	0.984	0.0104	0.00277
R2X (cum)	0.984	0.995	0.997

**Table 2 molecules-27-00025-t002:** Principal component factor loading for E–tongue data.

Taste	PC1	PC2	PC3
Sourness	0.391	0.146	−0.126
Bitterness	−0.163	0.512	0.772
Astringency	0.390	−0.065	0.228
Aftertaste-B	0.319	−0.344	0.443
Aftertaste-A	0.282	−0.438	0.135
Umami	−0.376	−0.209	0.197
Richness	0.382	0.131	0.169
Saltiness	−0.219	−0.579	0.214
Sweetness	−0.390	−0.077	0.093
R2X	0.682	0.216	0.058
R2X (cum)	0.682	0.899	0.956

**Table 3 molecules-27-00025-t003:** Key flavor components in PR samples with different processing levels.

Component	OT ^a^	Aroma Description ^b^	Relative Odor Activity Values (ROAV, %)										Change ^c^
			RPR	PPR1	PPR2	PPR3	PPR4	PPR5	PPR6	PPR7	PPR8	PPR9	
Dimethyl disulfide	7.6	Cabbage, onion, putrid	0.1	0.4	2.6	12.2	9.0	38.9	15.5	14.2	19.1	11.6	+
Hexanal	4.5	Apple, fat, green, oil	100	35.8	29.6	52.5	55.2	83.5	81.8	43.3	53.4	62.5	−
2-Heptanone	680	Cheese, fruity, green banana	0.1	0.02	0.02	0.02	0.02	0.02	0.02	0.01	0.01	0.00	−
Benzaldehyde	350	Almond, caramel	0.01	0.02	0.03	0.1	0.1	0.2	0.2	0.2	0.1	0.1	+
1-Octen-3-OL	1	Mushroom, earthy, fat, green	52.6	100	100	100	100	66.6	67.5	38.3	28.6	37.2	−
2-Pentylfuran	4.8	Butter, floral, fruit, green Bean	66.1	36.1	39.6	72.5	77.3	100	100	60.7	52.0	54.9	−/+/−
Nonanal	3.5	Fat, citrus, green	1.9	2.1	6.6	15.9	20.8	38.1	49.5	60.4	35.9	37.0	+
Beta-cyclocitral	5	Minty	0.3	0.6	0.7	2.4	2.0	5.8	4.0	2.3	1.9	3.0	+
m-Cresol	2	Medicinal, woody, leather, phenolic	ND	ND	ND	6.7	13.3	55.5	52.1	49.9	43.0	37.2	+
Furfural	100	Almond, bread, spice	ND	ND	ND	3.7	11.4	84.8	75.3	100	100	100	+
2-Furanmethanol	300	Burnt, caramel	ND	ND	ND	ND	ND	0.5	0.5	0.6	0.5	0.4	+
5-Methylfurfural	50	Almond, caramel	ND	ND	ND	ND	ND	2.0	2.0	5.1	8.0	7.8	+
Dimethyl trisulfide	3	Cabbage, fish, onion, sulfur	ND	0.7	3.5	8.2	6.7	15.9	6.6	5.0	4.2	3.0	+
Geraniol	40	Floral, sweet, rose, fruity, citrus	ND	0.2	0.1	0.2	ND	ND	ND	ND	ND	ND	+
Beta-caryophyllene	64	Wood, spice	ND	0.01	0.1	ND	ND	ND	ND	ND	ND	ND	+
Alpha-pinene	120	Cedarwood, pine, sharp	0.1	0.1	0.1	0.1	0.1	0.1	0.1	ND	ND	ND	NC/-

^a^ Odor threshold (OT) in ppb. ^b^ Aroma descriptors from online databases: FEMA (http://www.femaflavor.org, accessed on 10 July 2021) and Flavornet (http://www.flavornet.org, accessed on 10 July 2021). ^c^ Direction of change in ROAV for RPR vs. other processed PR samples: decrease (−), increase (+), or no change (NC). Only flavor compounds with ROAV ≥ 0.1 at least in PR samples are presented. ND, non-detectable.

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
