# Peer review of "Characterization, Classification, and Authentication of Polygonatum sibiricum Samples by Volatile Profiles and Flavor Properties"

_molecules, 2021, doi:10.3390/molecules27010025_

Round 1

Reviewer 1 Report

Nine cycles of steaming and drying are the processing method for Polygonatum sibiricum as a traditional Chinese medicine instead of food. The active components for the therapeutic effects of Polygonatum sibiricum are reported as polysaccharides, saponins, etc. The indicators to evaluate the traditional processing method should be the responsible components for the medicinal uses. Anyhow, the processing method for Polygonatum sibiricum is to improve the efficiency but not to improve the taste. 
The herbal medicine that can be used as food is mostly due to the biological function and most importantly, the safety, instead of the pleasant flavor. In fact, most of the root medicines are not so tasty. It is also not common to use Polygonatum sibiricum as the major ingredient of food. That is to say, the flavor of the food does not depend on Polygonatum sibiricum, because it is usually used as one of the ingredients with limited quantity. So does it really make sense to focus on the flavor? 
In this manuscript, a method used for food processing is used to evaluate a traditional processing method for traditional medicine. The therapeutic role of volatile compounds detected in the manuscript is not discussed. Thereby, the conclusion is not very scientific sound, although the method and results have been performed properly.
In addition, the conclusion is not enough to judge the processing method. Because anyhow, the major responsible active components are not evaluated. 
In 3.1, “After each cycle, samples were retained, and the color of the samples gradually became darker after each processing cycle (Figure S1)”. The description along with figures of samples are supposed to be in 2. Result and discussion. The number Figure S1 is supposed to appear before Figure S2 in the manuscript.
Page number 11 and 12 are missing

Reviewer 2 Report

Dear Authors,

I believe the manuscript requires major revisions:

  • Please check the English language grammar in the whole manuscript;
  • INTRODUCTION: the introduction is very limited, it is recommended to expand it, to better document the various treatments and the role of volatile compounds analysis coupled with chemometrics in quality control;
  • RESULTS and DISCUSSION:

Line 279-281: Hexanal is an indicator of rancidity, as affirmed by authors, but reporting a relative amount isn’t sufficient to show whether or not this is a problem? Line 233-338: There is no discussion of the volatiles variations in samples. Which metabolism? Is this expected?

  • TABLES S6-S8: I do not understand the significance of a standard deviation (std) of 0. Why do the authors obtain this result? Are the values of the replications equal? If so, the measurement error of the instrument used must be taken into account. Otherwise, the mean value and std have to be shown up to the first std significant digit;

Reviewer 3 Report

The work presented in the manuscript is of interest, but some corrections and further information should be added.

Lines 61-63: Unclear paragraph, please rewrite to clarify what is meant by   judgement index in this context.

Figure 6: The correspondence among the chromatograms of the fig 6, the relative content and the retention time in Tables S6, S7 and S8 is not clear. 
The individual chromatograms and mass spectra of the different compounds identified must be included in the Supporting information file. 
No information about the main peak at 5.6 min were added.

Figure 6: The correspondence among the chromatograms in figure 6, the relative content and retention time in tables S6, S7 and S8 is not clear. 
The individual chromatograms and mass spectra of the different compounds identified should be included in the Supporting Information file. 
No information on the main peak of approximately 5.6 minutes has been added.

Line 419. Clarify whether the 10 g are whole parts of pieces or powder. 100 g of water should be understood? What type of water, tap, distilled, MilliQ quality? Please clarify this.

Lines 473-475: The authors state that the identification of the different compounds has been done by comparing retention time and MoNA spectra. But no commercial standards have been used to compare retention times and spectra. In addition, the collection of MoNa spectra does not seem to be the most suitable for identifying the compounds analysed by a TRACE 1300 Series GC 458 coupled to a Thermo Fisher Scientific ISQ Series MS.

As indicated above, the authors should provide the chromatograms of the samples and the mass spectra of each compound identified in the Supporting Information file.

Round 2

Reviewer 1 Report

2.1. pH measurement
After each cycle, the color of the samples gradually became darker after each processing cycle (Figure S1).  

This sentence has a grammar problem, by repeating after each cycle.

Reviewer 2 Report

The authors responded correctly to the suggested revisions, improving the quality of the manuscript. Accept in present form

Reviewer 3 Report

The shortcomings and errors have been corrected, in my opinion the manuscript is worthy of publication.